# INTERACTION MAKES BETTER SEGMENTATION: AN INTERACTION-BASED FRAMEWORK FOR TEMPORAL ACTION SEGMENTATION

## ABSTRACT

Temporal action segmentation aims to classify the action category of each frame in untrimmed videos, primarily using RGB video and skeleton data. Most existing methods adopt a two-stage process: feature extraction and temporal modeling. However, we observe significant limitations in their spatio-temporal modeling: (i) Existing temporal modeling modules conduct frame-level and action-level interactions at a fixed temporal resolution, which over-smooths temporal features and leads to blurred action boundaries; (ii) Skeleton-based methods generally adopt temporal modeling modules originally designed for RGB video data, causing a misalignment between extracted features and temporal modeling modules. In this paper, we propose a novel **Inter**action-based framework for **Act**ion segmentation (**InterAct**) to address these issues. Firstly, we propose multi-scale frame-action interaction (MFAI) to facilitate frame-action interactions across varying temporal scales. This enhances the model's ability to capture complex temporal dynamics, producing more expressive temporal representations and alleviating the over-smoothing issue. Meanwhile, recognizing the complementary nature of different spatial modalities, we propose decoupled spatial modality interaction (DSMI). It decouples the modeling of spatial modalities and applies a deep fusion strategy to interactively integrate multi-scale spatial features. This results in more discriminative spatial features that are better aligned with the temporal modeling modules. Extensive experiments on six large-scale benchmarks demonstrate that InterAct significantly outperforms state-of-the-art methods on both RGB-based and skeleton-based datasets across diverse scenarios.

## 1 INTRODUCTION

Understanding human actions in videos is critical for various real-world applications, including surveillance (Luo et al., 2019), assistive rehabilitation (Filtjens et al., 2020), interactive robotics (Kenney et al., 2009), and virtual reality (Sudha et al., 2017). These applications require the analysis of long, untrimmed videos, which has motivated extensive research into the task of temporal action segmentation (TAS) (Farha & Gall, 2019; Li et al., 2021b; Ishikawa et al., 2021; Liu et al., 2022; Behrmann et al., 2022; Li et al., 2023a; Liu et al., 2023). The goal of TAS is to classify each video frame and segment videos into distinct, non-overlapping action segments.

Recent frame-action interaction strategies (Lu & Elhamifar, 2024) have achieved significant progress in this task. However, we observe that these methods tend to over-smooth temporal features, which in turn blur the boundaries between different action categories. Specifically, recognizing complex action sequences requires the integration of both long-term and short-term information to extract discriminative temporal features (Gao et al., 2021). Nevertheless, these methods rely solely on frame-action modeling at a fixed temporal resolution, as shown in Figure 1(a). This makes it difficult to capture temporal dependencies across varying time scales, resulting in over-smoothed temporal representations that blur action boundaries. Moreover, the iterative refinement process based on the fixed temporal resolution further amplifies this smoothing effect.

In the task of TAS, two primary types of data are commonly used: RGB video data (Kuehne et al., 2014) and skeleton data (Liu et al., 2017). However, existing skeleton-based methods (Filtjens et al.,

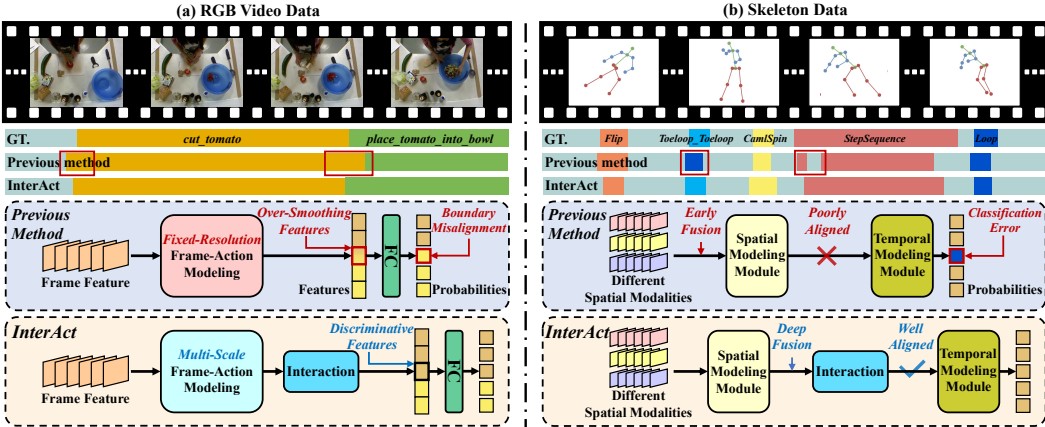

Figure 1: Comparison of existing methods and our proposed InterAct in TAS. (a) We introduce multi-scale frame-action modeling to enhance the interaction of temporal information, addressing the issues of over-smoothing features and boundary blurring caused by existing fixed-resolution modeling methods. (b) We apply a deep fusion strategy to decouple the modeling of spatial modalities and interactively integrate spatial information, overcoming the misalignment between extracted features and temporal modeling modules in existing skeleton-based methods.

2022; Xu et al., 2023; Tian et al., 2023) generally adopt modules originally designed for RGB video data during temporal modeling. This practice overlooks the differences in feature extraction between RGB video data and skeleton data. As a result, there is poor alignment between extracted features and temporal modeling modules. Specifically, RGB-based methods typically extract I3D features using pre-trained models (Carreira & Zisserman, 2017). These features are highly discriminative and effectively support temporal modeling. The design of temporal modeling modules in these methods heavily relies on these discriminative features. In contrast, as shown in Figure 1(b), previous skeleton-based methods (Filtjens et al., 2022; Xu et al., 2023; Li et al., 2023b) commonly adopt an early fusion strategy during feature extraction. Data from different spatial modalities are fused at the input stage before spatial modeling. This limits the model's ability to capture complex spatial dependencies, resulting in less discriminative spatial features. Consequently, these features misalign with the temporal modeling modules and result in classification errors.

To address the limitations mentioned above, we propose a novel Interaction-based framework for Action segmentation (InterAct). It comprises two core components: Multi-scale Frame-Action Interaction (MFAI) and Decoupled Spatial Modality Interaction (DSMI). Specifically, to avoid the effect of over-smoothing caused by iterative frame-action interaction at a fixed temporal resolution, MFAI introduces multiple temporal resolutions. By using various temporal scales ranging from coarse to fine granularity, MFAI performs temporal modeling simultaneously at both the frame and action levels. This facilitates interactions between the two, exploiting their complementary information to refine temporal representations. Particularly, frame-action interactions across different temporal scales focus on distinct temporal semantics. By enabling information transfer across these scales, MFAI learns more effective interaction patterns. As such, our InterAct can more comprehensively capture complex temporal dynamics in long action sequences. For skeleton data, inspired by the complementary nature of different spatial modalities, DSMI employs a deep fusion strategy. Initially, decoupled multi-scale spatial modeling is applied to data from different spatial modalities. The extracted multi-scale features are then fused interactively. By adopting DSMI, the more discriminative spatial features extracted can better capture the complex spatial relationships between joints, thereby aligning more effectively with the temporal modeling module.

Our main contributions are summarized as follows:

- For temporal modeling, we propose MFAI, which integrates multiple temporal resolutions for frame-action modeling, thereby enhancing temporal interactions. This module effectively captures complex temporal dependencies in long action sequences and performs well on both RGB video data and skeleton data.

- For skeleton data, we further propose a feature enhancement module DSMI. This module employs decoupled multi-scale spatial modeling for different spatial modalities. Through interactive fusion, the extracted spatial features become more discriminative and better aligned with the temporal modeling module.

- Extensive experimental results demonstrate that our proposed InterAct significantly outperforms existing state-of-the-art methods on both RGB-based and skeleton-based datasets.

## 2 RELATED WORKS

### 2.1 RGB-BASED TEMPORAL ACTION SEGMENTATION

Most temporal action segmentation works follow a similar two-stage process: feature extraction and temporal modeling. In RGB-based TAS, the first stage typically uses pre-trained model (Carreira & Zisserman, 2017) to extract I3D features from each video frame. Most research focuses on the design of the temporal modeling in the second stage. Existing temporal modeling methods can be categorized into three main types: frame-based methods, two-stage methods, and frame-action interaction methods. Frame-based methods model temporal dependencies between frames using temporal convolutional networks (Lea et al., 2017; Farha & Gall, 2019; Li et al., 2021b; Wang et al., 2020; Ishikawa et al., 2021; Singhania et al., 2023) or transformers (Yi et al., 2021; Bahrami et al., 2023). Although these approaches enhance temporal modeling through innovations such as multi-layer dilated convolutions (Farha & Gall, 2019; Li et al., 2020) and windowed attention (Yi et al., 2021), they still struggle to capture long-range dependencies. Recently, diffusion models (Liu et al., 2023) have also been applied to action segmentation, but leads to higher training and inference complexity. To better model long-range dependencies, the two-stage methods (Ahn & Lee, 2021; Behrmann et al., 2022; Jiang et al., 2023; Gan et al., 2024) recognize the significance of action-level modeling. These methods first learn initial frame features and predictions, then construct action features based on them and further refine the predictions. However, they fail to leverage the complementary information between the frame-level and action-level features. To address this limitation, frame-action interaction methods (Lu & Elhamifar, 2024) conduct temporal modeling at both the frame-level and action-level, enabling bidirectional information transfer between them. Nevertheless, these methods apply iterative frame-action modeling at a fixed temporal resolution, which over-smooths temporal features and limits the temporal modeling capability. To avoid over-smoothing temporal features, we propose multi-scale frame-action interaction (MFAI). It performs temporal modeling at both the frame and action level across multiple temporal resolutions. By facilitating the interactions of diverse temporal semantics, it generates more comprehensive temporal representations, thereby improving the model's capacity to capture complex temporal dynamics.

### 2.2 SKELETON-BASED TEMPORAL ACTION SEGMENTATION

In skeleton-based TAS, most works generally adopt frame-based methods from RGB-based approaches for temporal modeling. The primary focus is on designing feature extraction methods. Existing skeleton-based feature extraction methods can be divided into two categories: cascaded spatio-temporal modeling and decoupled spatio-temporal modeling. Cascaded spatio-temporal modeling methods conduct single or multiple cascaded spatio-temporal interactions to extract features. Early approaches, based on Farha & Gall (2019), replaced the initial stage of temporal convolutions with spatio-temporal graph convolutions (Filtjens et al., 2022) or spatio-temporal attention modules (Tian et al., 2023) to improve the ability to capture spatio-temporal features. To further refine spatial semantics, Liu et al. (2022) introduced spatial focus attention, while Tan et al. (2023) proposed a multi-branch transfer fusion module to model spatial dependencies. Similarly, Li et al. (2023a) enhanced spatio-temporal modeling by introducing an involving distinguished temporal graph convolution network. However, these cascaded spatio-temporal interactions tend to over-smooth the extracted features and fail to capture complex spatio-temporal information effectively. To mitigate this limitation, Li et al. (2023b) proposed a decoupled spatio-temporal modeling method. This method adopts unified spatial modeling to extract spatial sub-features, which then interact with temporal features, thereby avoiding cascaded spatio-temporal interactions. However, these methods commonly adopt an early fusion strategy, where data from different spatial modalities are combined at the input stage before spatial modeling. This diminishes the discriminative capacity of the extracted spatial features and hinders their alignment with the temporal modeling module. Indeed, different spatial modalities

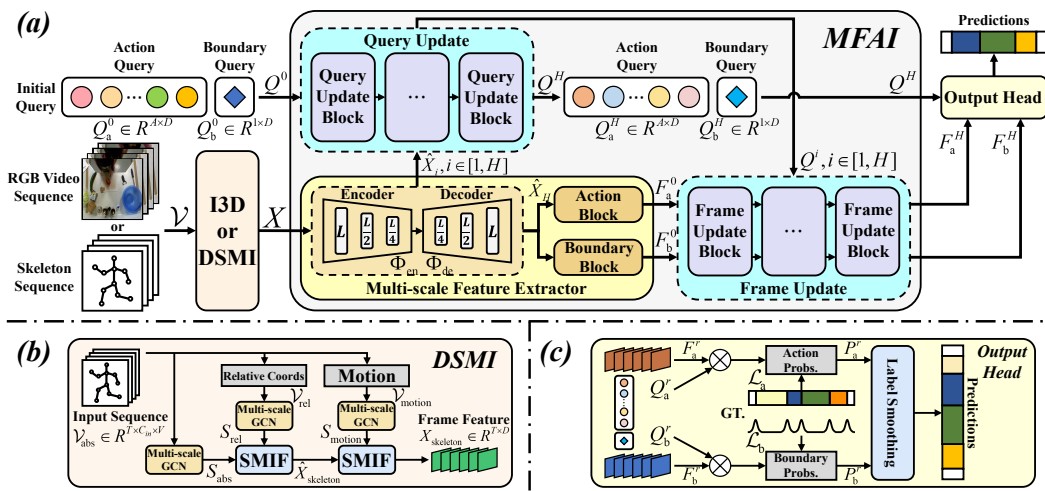

Figure 2: Framework of the proposed InterAct. First, a frame-level encoder is used to extract features. For RGB video data, we employ I3D, while for skeleton data, we use DSMI (illustrated in (b)) to capture discriminative spatial features. Then, MFAI (illustrated in (a)) models temporal dependencies. Finally, predictions are generated using the output head (illustrated in (c)).

contain rich complementary information. Motivated by this, we propose decoupled spatial modality interaction (DSMI), which applies a deep fusion strategy to decouple the modeling of different spatial modalities and integrate multi-scale spatial features interactively. As such, the extracted spatial features become more discriminative, providing better support for temporal modeling.

## 3 METHOD

In this section, we present the details of the proposed interaction-based framework, InterAct, for temporal action segmentation (TAS). In Sec. 3.1, we first introduce the tasks of RGB-based TAS and skeleton-based TAS, along with the pipeline of InterAct. Then, the decoupled spatial modality interaction (DSMI) and the multi-scale frame-action interaction (MFAI) are proposed in Sec. 3.2 and Sec. 3.3, respectively. Finally, we provide details of the loss functions in Sec. 3.4.

### 3.1 PROBLEM STATEMENT AND PIPELINE

Given a video with $T$ frames, our goal is to identify the category for each frame $Y = [y_1, \ldots, y_T] \in [1, \ldots, A]^T$, where $A$ is the total number of action classes. To achieve better segmentation performance, we propose a framework named InterAct, as shown in Figure. 2. In TAS, two primary types of data are commonly used: RGB video data and skeleton data. Due to the distinct characteristics of these data types, we employ separate frame-level encoders for feature extraction. For RGB video sequences $\mathcal{V}_{\text{RGB}} \in R^{T \times H \times W}$, following the previous (Farha & Gall, 2019; Liu et al., 2023), we extract I3D (Carreira & Zisserman, 2017) features $X_{\text{I3D}} \in R^{T \times C}$, where $H$, $W$ and $C$ represent the height, width and feature dimension, respectively. For skeleton sequences $\mathcal{V}_{\text{skeleton}} \in R^{T \times C_{in} \times V}$, we use the proposed DSMI as the frame-level encoder to extract more discriminative spatial features, denoted as $X_{\text{skeleton}} \in R^{T \times C}$, where $C_{in}$ is the input feature dimension. Next, we apply MFAI to both $X_{\text{I3D}}$ and $X_{\text{skeleton}}$ for multi-scale frame-action interaction and temporal modeling. Additionally, following Ishikawa et al. (2021), we introduce a boundary query to help the learning of action boundaries, which effectively reduces over-segmentation errors.

### 3.2 DECOUPLED SPATIAL MODALITY INTERACTION

As mentioned before, previous skeleton-based methods (Filtjens et al., 2022; Xu et al., 2023; Tian et al., 2023) suffers from misalignment between the extracted features and the temporal modeling

module. To address this issue, we propose a decoupled spatial modality interaction module (DSMI) to strengthen compatibility between them, as shown in Figure 2(b).

Specifically, we explore three distinct spatial modalities: absolute coordinates, relative coordinates, and motion, with the latter two derived from the absolute coordinates. The absolute coordinates represent the positional information of the human body, while the relative coordinates describe changes in joint positions relative to the body's center. The motion modality, on the other hand, captures the movement of joints across consecutive frames.

**Multi-scale Spatial Modeling.** We first apply multi-scale spatial modeling (Li et al., 2023b) to thoroughly exploit the spatial information embedded in these modalities. Taking the skeleton sequence of the absolute coordinate modality $\mathcal{V}_{\text{abs}} \in R^{T \times C_{in} \times V}$ as an example, we define a $k$-adjacency matrix $A^{(k)} \in R^{V \times V}$ to represent the physical connections between body joints:

$$A_{i,j}^{(k)} = \begin{cases} 1, & \text{if } d(\alpha_i, \alpha_j) = k, \\ 1, & \text{if } i = j, \\ 0, & \text{otherwise}, \end{cases} \tag{1}$$

where $d(\alpha_i, \alpha_j)$ denotes the shortest distance between joint $\alpha_i$ and $\alpha_j$. The dependencies between joints at distance $k$ can be captured via matrix multiplication $\mathcal{V}_{\text{abs}} A^{(k)}$. Additionally, a learnable adjacency matrix $B^{(k)} \in R^{V \times V}$ is introduced to adaptively learn the spatial dependencies between joints. By leveraging both adjacency matrices, the multi-scale spatial features of the absolute coordinate modality $S_{\text{abs}} \in R^{T \times C \times V}$ are aggregated as:

$$S_{\text{abs}} = \text{MLP}(W \mathcal{V}_{\text{abs}}([(\hat{A}^{(1)} + B^{(1)}) \| \cdots \| (\hat{A}^{(K)} + B^{(K)})])), \tag{2}$$

where $\|$ denotes the concatenation operation, $W$ is a weight tensor, $K$ is a model hyperparameter that controls the farthest distance, and $\hat{A}^{(k)}$ is the normalized adjacency matrix (Yan et al., 2018; Liu et al., 2020). The MLP (multi-layer perceptron) adjusts the feature dimensions. Similarly, we extract multi-scale spatial features for other modalities, i.e., $S_{\text{rel}} \in R^{T \times C \times V}$ and $S_{\text{motion}} \in R^{T \times C \times V}$.

**Spatial Modality Interactive Fusion.** We then leverage a deep fusion strategy to capture the complementary relationships between different spatial modalities, facilitating the fusion of multi-scale features. Using the multi-scale features of the input modality $S_{\text{abs}}$ as the reference, we model the correlations between it and other modalities sequentially. This progressive integration yields the final spatial feature $X_{\text{skeleton}} \in R^{T \times C}$. The process can be formally described as follows:

$$\begin{aligned} \hat{X}_{\text{skeleton}} &= \text{MLP}(\text{SAttn}([S_{\text{abs}} \| S_{\text{rel}}])), \\ X_{\text{skeleton}} &= \text{MLP}(\text{SAttn}([\hat{X}_{\text{skeleton}} \| S_{\text{motion}}])), \end{aligned} \tag{3}$$

where $\text{SAttn}$ denotes the self-attention layer. Through this interactive fusion, DSMI is able to extract more discriminative spatial features to better support temporal modeling.

### 3.3 MULTI-SCALE FRAME-ACTION INTERACTION

It is critical to capture rich temporal dependencies in long action sequences. However, we observe that existing frame-action interaction strategies tend to over-smooth temporal features, limiting their temporal modeling capabilities. To solve this problem, we propose a multi-scale frame-action interaction module (MFAI) that utilizes multiple temporal resolutions to enhance temporal interactions, as shown in Figure 2(a). The module consists of a multi-scale feature extractor and frame-action interactions at different temporal resolutions. Next, we describe each component in detail.

**Multi-scale Feature Extractor.** Following Singhania et al. (2023), we adopt an encoder-decoder architecture to capture frame-level information at different temporal scales. Let $X \in R^{T \times C}$ denote the input features, where $T$ is the number of frames and $C$ is the input feature dimensions. The encoder consists of six layers, denoted as $\{\Phi_{\text{en}}^{(u)} : u \leq 6\}$. The output of the $u$-th layer has dimensions $R^{T^{(u)} \times D}$, where $T^{(u)}$ is the temporal dimension at layer $u$ and $D$ is the feature dimensions. Each encoder layer applies a 1D depthwise convolution to halve the temporal dimension, i.e., $T^{(u)} = \lceil \frac{T}{2^u} \rceil$, followed by an asformer layer (Yi et al., 2021) to capture contextual information at the corresponding temporal scale. The decoder mirrors the encoder's structure, also comprising six layers, denoted as $\{\Phi_{\text{de}}^{(u)} : u \leq 6\}$. Each decoder layer includes an up-sampling unit and a convolution block with

the output having dimensions $R^{T^{(6-u)} \times D}$. For each $u$, the up-sampling unit linearly interpolates inputs to double the temporal length, and the output is fused with encoder $\Phi_{en}^{(6-u)}$'s output via skip connections. This fusion effectively integrates global and local information across multiple scales.

We utilize the outputs from the last $H$ layers of the decoder $\hat{X} = [\hat{X}_0, \ldots, \hat{X}_H]$ to construct and refine action-level features during the frame-action interaction step, where $\hat{X}_u \in R^{T^{(H-u)} \times D}$. The final output $\hat{X}_H$ is then passed through an action block (Li et al., 2023b) and a boundary block (Li et al., 2021b), generating the initial frame-level features $F^0 = [F_a^0, F_b^0]$. Here, $F_a^0 \in R^{T \times D}$ denotes the frame-level action features, and $F_b^0 \in R^{T \times D}$ denotes the frame-level boundary features.

**Frame-Action Interaction.** We progressively model frame-action interactions across multiple temporal scales, enhancing information transfer between frame-level and action-level features, as well as across different time scales. This allows the model to effectively integrate low-level detail from frame-level features with high-level dependencies from action-level features. Let $Q^0 = [Q_a^0, Q_b^0]$ denote the initial action-level features, where $Q_a^0 \in R^{A \times D}$ and $Q_b^0 \in R^{1 \times D}$ are action query and boundary query, respectively. Both are randomly initialized. The frame-action modeling at each temporal scale involves two steps: Query Update and Frame Update. The inputs to the first frame-action modeling stage are $(F^0, Q^0, \hat{X}_0)$, and its outputs are the refined features $(F^1, Q^1)$.

In the Query Update step, the initial action-level features $Q^0$ and the coarse-grained decoder output $\hat{X}_0$ are used to update the action-level features via the Query Update Block (QBlock). Each QBlock employs a transformer with self-attention to capture dependencies between action and boundary queries. Then, guided by the decoder output, we further refine the action-level features using a frame-to-action cross-attention layer, where $Q^0$ serves as Query and $\hat{X}_0$ as Key and Value:

$$
\begin{aligned}
Q^1 &= \text{QBlock}(Q^0, \hat{X}_0), \\
&= \text{MLP}(\text{CAttn}(\text{SAttn}(Q_a^0, Q_b^0), \hat{X}_0)).
\end{aligned}
\tag{4}
$$

Here, SAttn and CAttn denote the self-attention and cross-attention layers, respectively. MLP is used to adjust the feature dimensions. The updated action-level features are more sensitive to both action categories and boundary information.

In the Frame Update step, based on the updated action-level features $Q^1$ and the initial frame-level features $F^0$, frame-level features are refined through the Frame Update Block (FBlock). Each FBlock refines both frame-level action and boundary features using an action-to-frame cross-attention layer. In this step, $F_a^0$ and $F_b^0$ are treated as Query, while $Q^1$ serves as Key and Value:

$$
\begin{aligned}
F^1 = [F_a^1, F_b^1] &= \text{FBlock}(F^0, Q^1), \\
&= [\text{CAttn}(F_a^0, Q^1), \text{CAttn}(F_b^0, Q^1)].
\end{aligned}
\tag{5}
$$

Lastly, we pass the updated features $(Q^1, F^1)$ and the finer-grained decoder output $\hat{X}_1$ to the next frame-action modeling stage. This process is repeated iteratively until we obtain $(Q^H, F^H)$ from the final stage, where $H$ is the total number of stages.

**Generating Predictions.** As shown in Figure 2(c), we use the output $(Q^r, F^r)$ from each time scale $r$ to predict the probability for the action category and action boundary of each frame. Specifically, the action probability $P_a^r \in R^{A \times T}$ is obtained by computing the dot product between the action query $Q_a^r$ and the frame-level action features $F_a^r$. The boundary probability $P_b^r \in R^{1 \times T}$ is obtained in a similar way. During inference, based on the final probabilities $P_a^H$ and $P_b^H$, we apply the label smoothing strategy (Ishikawa et al., 2021) to generate the final predictions.

### 3.4 LOSS FUNCTION

We optimize the action probability $P_a^r$ using both frame-level and action-level losses, following Gan et al. (2024). Specifically, we apply focal loss (Ross & Dollár, 2017) for frame-level classification and dice loss (Milletari et al., 2016) at the action-level to better capture the temporal distribution of each action category. For a video with $N$ action categories, we first generate the temporal mask label $M = [M_1, \ldots, M_N] \in R^{N \times T}$ based on the label $Y$, where $M_i$ is defined as:

$$
M_i = [m_1, \ldots, m_T], m_t = \begin{cases} 1, & \text{if } y_t = i, \\ 0, & \text{else.} \end{cases}
\tag{6}
$$

Here, $y_t$ represents the label of the $t$-th frame. We then extract the corresponding $N$ classes from $P_a^r$, denoted as $P_a^r(M) \in R^{N \times T}$. Overall, the loss function for action probability is formulated as:

$$\mathcal{L}_a = \sum_{r=1}^{H} [\lambda_{\text{focal}} \mathcal{L}_{\text{focal}}(Y, P_a^r) + \lambda_{\text{dice}} \mathcal{L}_{\text{dice}}(M, P_a^r(M))], \tag{7}$$

where $\lambda_{\text{focal}}$ and $\lambda_{\text{dice}}$ are the weights for the focal loss and dice loss, respectively. For boundary probability $P_b^r$, we use a binary logistic regression loss $\mathcal{L}_b$ at each stage as follow:

$$\mathcal{L}_b = \sum_{r=1}^{H} \frac{1}{T} \sum_{t=1}^{T} [g(Y_b(t)) \cdot \log P_b^r(t) + (1 - g(Y_b(t)) \cdot \log(1 - P_b^r(t)], \tag{8}$$

where $Y_b(t)$ is the ground truth that takes the value of 1 at action boundaries, and $g(\cdot)$ denotes a Gaussian filter used to smooth boundaries. In summary, the action probabilities and the boundary probabilities are jointly trained with the following loss function:

$$\mathcal{L} = \mathcal{L}_a + \gamma \mathcal{L}_b \tag{9}$$

where $\gamma$ is a hyperparameter that balances the contributions of the two losses.

## 4 EXPERIMENT

### 4.1 DATASET

We evaluate the proposed InterAct for action segmentation on six challenging datasets covering various test scenarios. These include daily cooking activities (e.g., Breakfast (Kuehne et al., 2014) and 50Salads (Stein & McKenna, 2013)), competitive sports (e.g., MCFS-22 (Liu et al., 2021) and MCFS-130 (Liu et al., 2021)), daily activities (e.g., PKU-MMD (Liu et al., 2017)), and typical warehousing activities (e.g., LARa (Niemann et al., 2020)).

**Breakfast** consists of 1712 third-person view videos with 48 distinct actions related to breakfast preparation. **50Salads** includes 50 videos in which 25 participants prepare two types of mixed salads. It contains 17 action classes recorded from a top-down view. **MCFS-22** is a high-quality action segmentation dataset with 271 long sequences of skeleton-based actions, totaling over 1.73 million frames. The actions are categorized into 22 classes. **MCFS-130** features more fine-grained actions in both spatial and temporal dimensions compared to MCFS-22, covering 130 action categories. It provides two types of data: RGB video data and skeleton data. **PKU-MMD** is a large-scale human action understanding dataset. It contains 1009 long continuous sequences of 52 distinct actions, recorded from three camera views with 13 subjects. Following Li et al. (2023b), we use two evaluation protocols: cross-subject (X-sub) and cross-view (X-view). **LARa** is a continuous action dataset involving 14 participants performing typical warehousing activities. It consist of 377 long videos covering 8 action classes, captured in 3 different real-world warehousing scenarios.

### 4.2 EVALUATION METRICS

Following previous works, we report three evaluation metrics, i.e., frame-wise accuracy (Acc), segmental edit score (Edit), and segmental F1 scores with overlapping thresholds of 10%, 25% and 50%, denoted as $F1@\{10, 25, 50\}$. We perform 4-fold cross-validation on Breakfast, 5-fold cross-validation on 50Salads, MCFS-22, and MCFS-130. For PKU-MMD (X-sub), PKU-MMD (X-view), and LARa, we use single validation for evaluation.

### 4.3 IMPLEMENTATION DETAILS

For DSMI, we set $K = 13$ following Li et al. (2023b). For MFAI, the number of stages for frame-action modeling $H$ is set to 3 (we discuss the impact of the number of $H$ in the Appendix). The loss function parameters are configured as $\lambda_{focal} = \lambda_{dice} = \gamma = 1$. We use AdamW optimizer and a cosine learning rate schedule for training on all datasets. The initial learning rate is 5e-4 for PKU-MMD and 1e-4 for other datasets. For Breakfast and PKU-MMD, we train for 66 epochs, using batch sizes of 1 and 4, respectively. For MCFS-22 and MCFS-130, we train for 132 epochs, using a batch size of 1. For 50Salads, we train for 212 epoch, using a batch size of 1. All experiments in the comparison study use the above setting and are conducted on a single RTX 4090 GPU.

Table 1: Comparison with the state-of-the-art on Breakfast, 50Salads and MCFS-130 (RGB). The underlined results represent the reproduced results.

| Method | Breakfast | | | | | 50Salads | | | | | MCFS-130 (RGB) | | | | |
|---|---|---|---|---|---|---|---|---|---|---|---|---|---|---|---|
| | F1@{10,25,50} | | | Edit | Acc | F1@{10,25,50} | | | Edit | Acc | F1@{10,25,50} | | | Edit | Acc |
| MS-TCN | 52.6 | 48.1 | 37.9 | 61.7 | 66.3 | 76.3 | 74.0 | 64.5 | 67.9 | 80.7 | 36.6 | 30.5 | 20.0 | 36.3 | 58.0 |
| BCN | 68.7 | 65.5 | 55.0 | 66.2 | 70.4 | 82.3 | 81.3 | 74.0 | 74.3 | 84.4 | - | - | - | - | - |
| C2F-TCN | 72.2 | 68.7 | 57.6 | 69.6 | 76.0 | 84.3 | 81.8 | 72.6 | 76.4 | 84.9 | - | - | - | - | - |
| ASRF | 74.3 | 68.9 | 56.1 | 72.4 | 67.6 | 84.9 | 83.5 | 77.3 | 79.3 | 84.5 | 45.4 | 40.1 | 27.9 | 47.1 | 55.0 |
| ETSN | 74.0 | 69.0 | 56.2 | 70.3 | 67.8 | 85.2 | 83.9 | 75.4 | 78.8 | 82.0 | 38.7 | 33.0 | 21.1 | 47.0 | 58.1 |
| ASFormer | 76.0 | 70.6 | 57.4 | 75.0 | 73.5 | 85.1 | 83.4 | 76.0 | 79.6 | 85.6 | 37.5 | 32.6 | 22.5 | 36.1 | 57.6 |
| UVAST | 76.9 | 71.5 | 58.0 | 77.1 | 69.7 | 89.1 | 87.6 | 81.7 | 83.9 | 87.4 | - | - | - | - | - |
| RTK | 76.9 | 72.4 | 60.5 | 76.1 | 73.3 | 87.4 | 86.1 | 79.5 | 81.4 | 85.9 | - | - | - | - | - |
| LTContext | 77.6 | 72.6 | 60.1 | 77.0 | 74.2 | 89.4 | 87.7 | 82.0 | 83.2 | 87.7 | - | - | - | - | - |
| DiffAct | 80.3 | 75.9 | 64.6 | 78.4 | 76.4 | 90.1 | 89.2 | 83.7 | 85.0 | 88.9 | - | - | - | - | - |
| DSTN | 80.4 | 75.7 | 64.7 | 78.0 | 73.7 | - | - | - | - | - | 53.4 | 48.8 | 36.5 | 57.1 | 58.8 |
| ASQuery | 80.7 | 76.5 | 66.5 | 78.4 | 77.9 | 88.6 | 87.9 | 83.6 | 84.0 | 88.2 | 52.7 | 48.6 | 38.8 | 49.2 | 61.1 |
| FACT | 81.4 | 76.5 | 66.2 | 79.7 | 76.2 | 87.1 | 85.7 | 80.3 | 81.3 | 86.6 | 44.0 | 38.1 | 31.9 | 34.8 | 59.5 |
| InterAct (Ours) | **82.3** | **78.1** | **68.3** | **80.2** | **78.2** | **90.2** | **89.5** | **85.3** | **85.5** | **88.9** | **58.9** | **55.4** | **45.9** | **57.3** | **62.2** |

Table 2: Comparison with the state-of-the-art on LARa and PKU-MMD using the benchmark of X-sub and X-view. The underlined results represent the reproduced results. * indicates the MFAI module is applied directly for temporal modeling of the input without feature extraction.

| Method | PKU-MMD (X-sub) | | | | | PKU-MMD (X-view) | | | | | LARa | | | | |
|---|---|---|---|---|---|---|---|---|---|---|---|---|---|---|---|
| | F1@{10,25,50} | | | Edit | Acc | F1@{10,25,50} | | | Edit | Acc | F1@{10,25,50} | | | Edit | Acc |
| MS-TCN | 63.4 | 60.2 | 54.2 | 66.4 | 65.5 | 58.6 | 53.6 | 39.4 | 56.6 | 58.2 | 52.4 | 45.7 | 39.6 | 44.2 | 65.8 |
| FACT | 76.0 | 71.4 | 56.6 | 72.1 | 69.6 | 71.7 | 67.0 | 54.1 | 68.7 | 68.1 | 64.3 | 60.7 | 48.3 | 60.0 | 69.8 |
| MS-GCN | - | - | 51.6 | - | 68.5 | 61.3 | 56.7 | 44.1 | 58.1 | 65.3 | - | - | 43.6 | - | 65.6 |
| CTC | 69.9 | 66.4 | 53.8 | - | 69.2 | - | - | - | - | - | - | - | - | - | - |
| DeST | 74.5 | 71.0 | 58.7 | 69.3 | 70.3 | 69.3 | 65.6 | 52.0 | 64.7 | 67.3 | 70.3 | 68.0 | 57.7 | 64.2 | 75.1 |
| InterAct* (Ours) | 82.8 | 80.4 | 69.9 | 77.1 | 77.9 | 76.9 | 73.8 | 62.6 | 70.8 | 73.5 | 71.8 | 69.1 | 58.2 | 64.8 | 74.7 |
| InterAct (Ours) | **83.5** | **81.7** | **73.0** | **78.2** | **79.9** | **77.5** | **74.4** | **64.3** | **71.3** | **73.7** | **72.7** | **70.1** | **59.5** | **65.6** | **75.9** |

## 4.4 COMPARISON WITH THE STATE-OF-THE-ART

To verify the effectiveness of our approach, we compare the proposed InterAct with state-of-the-art methods on six datasets, including RGB-based datasets (e.g. Breakfast, 50 salads, MCFS-130) and skeleton-based datasets (e.g. MCFS-22, MCFS-130, PKU-MMD, LARa). The results are presented in Tables 1, 2, and 3.

**Compared with RGB-based Methods.** We first compare InterAct against RGB-based methods (e.g., Farha & Gall (2019); Liu et al. (2023); Lu & Elhamifar (2024)) on RGB-based datasets. Unlike InterAct, these methods fail to fully leverage the complementary information between frame-level and action-level features, thus limiting their performance. As shown in Table 1, InterAct consistently outperforms these methods. For example, on Breakfast in terms of F1@50, InterAct surpasses the frame-based method DiffAct by 3.7%, the two-stage method ASQuery by 1.8%, and the frame-action interaction method FACT by 2.1%.

**Compared with skeleton-based Methods.** We then compare InterAct with skeleton-based methods (e.g., Filtjens et al. (2022); Li et al. (2023b)) on skeleton-based datasets. Although these methods incorporate spatio-temporal modeling, their spatial features are not well-aligned with the temporal module, which hampers their performance. As shown in Table 2 and 3, InterAct consistently achieves the best performance, especially on F1 scores. For example, in terms of F1@50, InterAct outperforms the previous SOTA method DeST by 14.3% on PKU-MMD (X-sub), and by 1.3% on challenging MCFS-130. Notably, even when MFAI is applied directly for temporal modeling of the input sequence, InterAct achieves state-of-the-art or competitive performance on these datasets.

## 4.5 ANALYSIS OF DECOUPLED SPATIAL MODALITY INTERACTION

**Different strategies of spatial feature extraction.** To validate the effectiveness of the proposed DSMI module, we compare different spatial feature extraction strategies. Using the scenario without spatial modeling as the baseline, we compare the existing strategy (Filtjens et al., 2022; Liu et al., 2022; Li et al., 2023a) (i.e., Concatenation) with DSMI. As shown in Table 4, 1) Spatial modeling significantly improves performance compared to the baseline, underscoring its importance

Table 3: Comparison with the state-of-the-art on MCFS-22 and MCFS-130. The underlined results represent the reproduced results. * indicates the MFAI module is applied directly for temporal modeling of the input without feature extraction.

| Method | MCFS-130 (Skeleton) | | | | | MCFS-22 | | | | |
|---|---|---|---|---|---|---|---|---|---|---|
| | F1@{10,25,50} | | | Edit | Acc | F1@{10,25,50} | | | Edit | Acc |
| MS-TCN | 56.4 | 52.2 | 42.5 | 54.5 | 65.7 | 74.3 | 69.7 | 59.5 | 74.2 | 75.6 |
| ASRF | 66.7 | 62.3 | 51.9 | 65.6 | 65.6 | 83.3 | 80.1 | 69.2 | 77.3 | 75.5 |
| ASFormer | 68.3 | 64.0 | 55.1 | 69.1 | 67.5 | 82.8 | 77.9 | 66.9 | 82.3 | 78.7 |
| FACT | 71.4 | 67.7 | 57.2 | 72.6 | 68.6 | 79.3 | 75.1 | 64.1 | 80.2 | 76.6 |
| MS-GCN | 52.4 | 48.8 | 39.1 | 52.6 | 64.9 | 75.7 | 70.5 | 57.9 | 72.6 | 75.5 |
| SFA+MS-TCN | - | - | - | - | - | 81.3 | 77.4 | 67.0 | 80.0 | 80.7 |
| IDT-GCN | 70.7 | 67.3 | 58.6 | 70.2 | 68.6 | 88.0 | 84.9 | 74.9 | 84.5 | 79.9 |
| DeST | 79.0 | 75.4 | 66.0 | 78.4 | 73.1 | 88.1 | 85.4 | 76.2 | 84.9 | 80.5 |
| InterAct* (Ours) | 79.1 | 75.3 | 66.4 | 78.3 | 72.4 | 88.6 | 85.0 | 74.9 | 87.2 | 80.7 |
| InterAct (Ours) | **80.1** | **76.4** | **67.6** | **78.5** | **73.4** | **89.7** | **86.1** | **76.5** | **88.5** | **81.7** |

Table 4: Comparison of various spatial feature extraction strategies on the PKU-MMD (X-sub).

| Method | F1@{10,25,50} | | | Edit | Acc |
|---|---|---|---|---|---|
| Baseline | 82.8 | 80.4 | 69.9 | 77.1 | 77.9 |
| Concatenation | 83.1 | 81.1 | **73.0** | 77.4 | 79.3 |
| InterAct (Ours) | **83.5** | **81.7** | **73.0** | **78.2** | **79.9** |

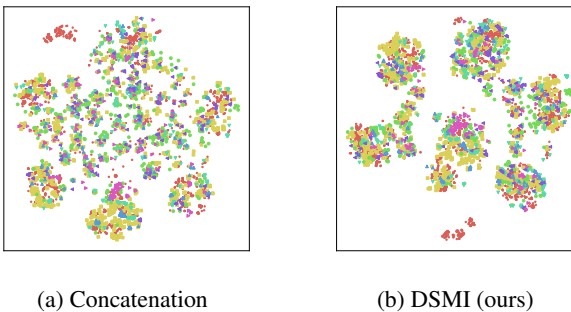

(a) Concatenation        (b) DSMI (ours)

Figure 3: The t-SNE visualization of spatial features extracted by existing methods (i.e., Concatenation) and DSMI. Different colors indicate the different categories in the MCFS-22. The spatial features we extract are more discriminative and better align with the temporal modeling module.

in skeleton-based TAS. 2) DSMI consistently outperforms Concatenation for all evaluation metrics. This demonstrates its ability to capture complex spatial dependencies by effectively leveraging complementary information from different spatial modalities.

**Qualitative analysis.** We further provide qualitative results for a more comprehensive analysis. We use t-SNE (Van der Maaten & Hinton, 2008) to visualize the spatial features extracted by existing methods (Li et al., 2023b) and DSMI. As shown in Figure 3, the spatial features produced by DSMI exhibit significantly better discriminative properties compared to those from existing methods. This enables better alignment with the temporal module and provides stronger support for temporal modeling. However, since DSMI models spatial dependencies based on skeletal poses from a limited number of local frames, spatial features of action categories with similar poses may still converge. In such cases, temporal dependencies between frames are needed for further differentiation.

### 4.6 ANALYSIS OF MULTI-SCALE FRAME-ACTION TEMPORAL INTERACTION

**Different strategies of frame-action interaction.** Here, we evaluate the proposed multi-scale frame-action interaction (MFAI) strategy against the previous method FACT (Lu & Elhamifar, 2024) to verify its superiority. As shown in Table 1, 2 and 3, MFAI consistently achieves better performance. These results indicate that MFAI predicts more accurate and complete action segments

Table 5: Abalation result for the proposed modules on the MCFS-130 dataset (split #1).

| Baseline | DSMI | MFAI | F1@{10,25,50} | | | Edit | Acc |
|---|---|---|---|---|---|---|---|
| ✓ | | | 77.1 | 73.5 | 64.7 | 76.3 | 71.5 |
| ✓ | ✓ | | 79.2 | 75.6 | 66.3 | 77.5 | 71.8 |
| ✓ | | ✓ | 79.1 | 75.3 | 66.4 | 78.3 | 72.4 |
| ✓ | ✓ | ✓ | **80.2** | **76.5** | **67.0** | **78.4** | **72.7** |

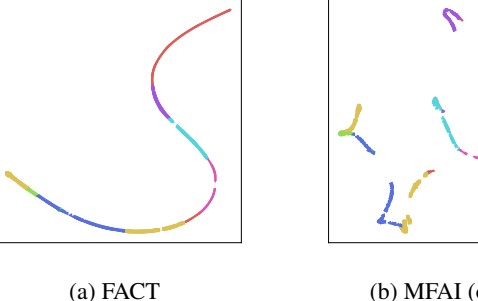

(a) FACT                    (b) MFAI (ours)

Figure 4: Visualization of temporal feature embeddings generated by FACT and MFAI. Different colors indicate the different categories in the PKU-MMD (X-sub). Compared to FACT, our temporal features exhibit more distinct category boundaries, mitigating the effects of over-smoothing.

across various scenes. This improvement is attributed to MFAI's enhanced capacity to capture richer temporal semantic information within action sequences.

**Qualitative analysis.** To delve deeper into the differences between FACT and MFAI and their impact on performance, we visualize the embedding distribution of the temporal features generated by both methods. As shown in Figure 4, FACT's frame-action interaction strategy results in a convergence of feature representations between different categories, making action boundaries ambiguous. This shows that modeling frame-action interactions at a fixed temporal resolution is prone to the problem of over-smoothing features. In contrast, by incorporating multi-scale frame-action interaction, our proposed MAFI mitigates this issue and yields more distinct category boundaries.

### 4.7 ABLATION STUDIES

**Effect of each proposed module.** To inspect the impact of the proposed spatial module DSMI and temporal module MFAI, a set of comparative experiments with different module combinations are conducted in Table 5. In the baseline setting, the model does not employ spatial modeling and relies solely on frame-level temporal modeling. It is observed that DSMI and MFAI are inherently strong spatial and temporal feature extractors, respectively. Moreover, these two modules are well-adapted and complement each other effectively. By combining them, we can achieve the best performance.

## 5 CONCLUSION

In this paper, we propose a novel framework InterAct for temporal action segmentation (TAS). Unlike previous frame-action interaction approaches, InterAct incorporates multiple temporal resolutions. It performs frame-action interactions across different temporal scales. This design effectively captures temporal semantic information and mitigates the over-smoothing issues associated with fixed resolution modeling. It shows excellent performance on both RGB video data and skeleton data. Additionally, to address the misalignment between spatial features and temporal modules in skeleton-based TAS, we decouple different spatial modalities and apply a deep fusion strategy for adaptive inter-modal interactions. This approach extracts more discriminative spatial features to better support temporal modeling. Our method outperforms all state-of-the-art RGB-based and skeleton-based methods on six large-scale benchmark datasets across various scenarios. We believe that InterAct offers a unique and innovative perspective for addressing the challenges in TAS.

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

# A   APPENDIX

In the Appendix, we have added some experiments and detailed explanations mentioned in the main text. In section A.1, we discuss the impact of the number of temporal scales. In section A.2, we provide more detailed experimental results to verify the effectiveness of the proposed multi-scale frame-action interaction module (MFAI). In section A.3, we list some limitations of our framework and future development directions.

## A.1   IMPACT OF THE NUMBER OF $H$

This hyperparameter controls the number of temporal scales at which frame-action interactions occur within the MFAI. As shown in Table 6, increasing $H$ from 0 to 3 gradually improves performance. This indicates that incorporating temporal semantic information from multiple scales is beneficial. However, further increasing $H$ from 3 to 6 leads to a decline in performance. This is likely due to the excessively large temporal down-sampling rate during initial interactions. This results in a severe loss of local details and reduces the differences between frames, thus impairing classification accuracy.

Table 6: The impact of the number of stages for frame-action modeling ($H$) on the MCFS-130 dataset (split #1). $H = i$ indicates that frame-action modeling is performed $i$ times, and the predictions are derived from the action-level features $Q^i = [Q_{\mathrm{a}}^i, Q_{\mathrm{b}}^i]$ and the frame-level features $F^i = [F_{\mathrm{a}}^i, F_{\mathrm{b}}^i]$.

| $H$ | F1@{10,25,50} | | | Edit | Acc |
|---|---|---|---|---|---|
| 0 | 78.3 | 74.4 | 64.8 | 76.2 | 71.5 |
| 1 | 79.4 | 75.5 | 66.0 | 77.8 | 71.4 |
| 2 | 79.2 | 75.9 | 66.4 | 77.6 | 71.8 |
| 3 | **80.2** | **76.5** | **67.0** | **78.4** | **72.7** |
| 4 | 78.6 | 75.2 | 65.8 | 77.8 | 71.5 |
| 5 | 78.3 | 75.1 | 66.5 | 77.4 | 71.1 |
| 6 | 78.5 | 74.7 | 65.1 | 76.7 | 71.1 |

Table 7: The results of the proposed InterAct at different stages of frame-action modeling on the MCFS-130 dataset (split #1). $N_i = i$ denotes that the predictions are derived from the action-level features $Q^i = [Q_{\mathrm{a}}^i, Q_{\mathrm{b}}^i]$ and the frame-level features $F^i = [F_{\mathrm{a}}^i, F_{\mathrm{b}}^i]$.

| $N_i$ | F1@{10,25,50} | | | Edit | Acc |
|---|---|---|---|---|---|
| 0 | 74.0 | 69.8 | 61.8 | 70.3 | 68.9 |
| 1 | 79.2 | 75.4 | 66.6 | 77.5 | 71.7 |
| 2 | 79.3 | 75.6 | 66.5 | 78.0 | 72.1 |
| 3 | **80.2** | **76.5** | **67.0** | **78.4** | **72.7** |

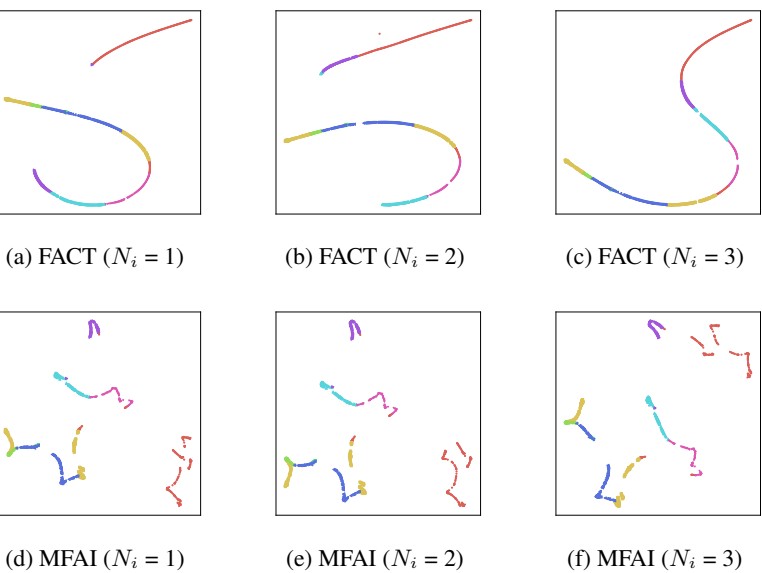

(a) FACT ($N_i = 1$)     (b) FACT ($N_i = 2$)     (c) FACT ($N_i = 3$)

(d) MFAI ($N_i = 1$)     (e) MFAI ($N_i = 2$)     (f) MFAI ($N_i = 3$)

Figure 5: Visualization of temporal feature embeddings generated by FACT and MFAI at different stages. Different colors indicate the different categories in the PKU-MMD (X-sub). FACT relies solely on iterative frame-action modeling with a fixed temporal resolution, resulting in temporal features that tend to be smooth. In contrast, with the introduction of multi-scale frame-action modeling, our temporal features exhibit more distinct category boundaries.

## A.2 MORE RESULTS COMPARING FRAME-ACTION INTERACTION STRATEGIES

In this section, we present additional quantitative and qualitative experiments on the multi-scale frame-action interaction (MFAI) for a more comprehensive analysis.

First, we conduct a more detailed analysis of the comparative results in Figure 4 and provide visualizations of the temporal feature embeddings generated by FACT and MFAI at different stages.

As shown in Figure 5, the frame-action interaction strategy employed by FACT leads to over-smoothing temporal features, which blurs the boundaries between various action categories. This over-smoothing effect intensifies as the model undergoes further iterations. In contrast, by incorporating multi-scale temporal semantic information, our proposed MFAI alleviates the issue of over-smoothing and achieves more separable category boundaries. Notably, after the first refinement stage, the temporal features generated by MFAI are already able to effectively distinguish different action categories. This is attributed to the successful integration of action-level features $Q^1 = [Q_a^1, Q_b^1]$ and frame-level features $F^1 = [F_a^1, F_b^1]$ during the first refinement stage. Specifically, the action-level features $Q^1$ capture long-range dependencies within the action sequence, guided by the coarse-grained decoder output $\hat{X}_0$. Simultaneously, the frame-level features $F^1$, produced by the multi-scale feature extractor and further refined with the assistance of $Q^1$, enrich the semantic information of temporal details. As a result, MFAI's output in the first stage significantly outperforms that of FACT and continues to improve through subsequent refinement iterations. However, as observed in the visualizations, the subsequent refinements appear less prominent, primarily because they focus on the improvement of specific local details.

Similarly, as shown in Table 7, the performance of MFAI steadily improves as frame-action interaction modeling progresses from coarse to fine granularity. This improvement is driven by the incorporation of more comprehensive temporal semantic information, enabling the model to more precisely capture and distinguish subtle variations between action categories. This, in turn, enhances the model's ability to recognize complex action sequences. However, the performance improvements from later refinement stages manifest less significantly than those achieved during the first refinement stage.

### A.3 LIMITATION AND FUTURE WORK

While InterAct significantly boosts the accuracy of fine-grained action segmentation, it still relies on costly frame-wise label annotations. To release this limitation, we plan to explore self-supervised long sequence modeling using methods such as contrastive learning (Li et al., 2021a; Mao et al., 2022; Zhou et al., 2023) to achieve better pre-trained models.

