# OpenReview forum: "Interaction Makes Better Segmentation: An Interaction-based Framework for Temporal Action Segmentation"
_ICLR.cc/2025/Conference — Submitted to ICLR 2025_

### Official Review · Reviewer_iX6Z · 2024-10-30

**Soundness:** 2
**Presentation:** 2
**Contribution:** 2
**Rating:** 5
**Confidence:** 5

**Summary:**

This paper introduces **InterAct**, a framework that advances temporal action segmentation by enhancing frame-action interactions and aligning spatial modalities in RGB and skeleton data. It targets limitations in existing temporal modeling techniques that over-smooth temporal features and blur action boundaries. The framework's two key components are Multi-scale Frame-Action Interaction (MFAI) and Decoupled Spatial Modality Interaction (DSMI). MFAI enables frame-action interaction at various temporal scales, and DSMI decouples spatial modeling for RGB and skeleton data to create better-aligned spatial representations. Extensive experimentation demonstrates InterAct’s effectiveness in improving state-of-the-art results across multiple datasets.

**Strengths:**

The idea of using a shared backbone for both skeleton-based and RGB-feature action segmentation is promising. The method is evaluated on six diverse datasets, showing substantial improvements over competitive benchmarks.

**Weaknesses:**

### 1. Unsupported Claims on Over-Smoothing and Blurred Boundaries
 The authors assert that RGB-based TAS methods suffer from over-smoothing, which they argue leads to blurred action boundaries. However, this claim lacks convincing evidence:
- More thorough proof is necessary to validate that over-smoothing indeed causes boundary detection issues.
- It is unclear how the results presented in Figure 4 support the claim of over-smoothing. Further clarification is needed to explain the causal link between the observed effects in Figure 4 and the issue of over-smoothing.
- Table 6 shows that three stages of frame-action modeling yield optimal results, yet the authors do not adequately explain why this stage limit would prevent oversmoothing, despite assuming it as an issue.

### 2. Limited Novelty and Borrowed Ideas
The paper presents limited novelty, appearing to draw heavily on ideas from prior works. Notably:
- The use of multiple temporal resolutions within an encoder-decoder model, along with multi-scale outputs for prediction, has already been explored in previous work (Singhania et al., 2023).
- Similarly, the implementation of segment-level learnable queries and cross-attention closely resembles techniques described in Behrmann et al. (2022).
The authors should clarify how their approach differs from or improves upon the multi-scale techniques used by Singhania et al. (2023) or the query-based methods proposed by Behrmann et al. (2022).
### 3. Inadequate Explanation on Addressing Spatial and Temporal Misalignment
The paper claims to address the misalignment between spatial features and temporal modules in skeleton-based TAS, yet it does not provide a satisfactory explanation of how the proposed method mitigates this issue. Specific details on how the approach prevents or corrects spatial-temporal misalignment would strengthen the paper’s argument.

### 4. Inconsistent Ablation Study and Limited Scope
The ablations presented in Tables 4 and 5 are conducted on different datasets, which weakens the comparison. Additional issues include:
- Table 5 appears to focus on skeleton data, suggesting that there is no ablation specifically on RGB-based action segmentation. It is necessary to see ablation results on the Breakfast dataset, a large RGB action segmentation dataset, to assess the robustness of the proposed method.
### 5. Missing Benchmark Results
In Table 1, results are missing for MCFS-130 on the DiffAct, UVAST, and LTContext benchmarks, despite the authors of these approaches making code available. The inclusion of these results would provide a more complete comparison.

### 6. Missing Ablation for Encoder and Decoder Layers
An ablation on the number of encoder and decoder layers is missing, which would be important to evaluate. Layer depth directly influences the granularity of temporal resolution, a factor critical to the proposed framework.

**Questions:**

Several important ablations and experiments are missing; see weaknesses for further details.

---

> ### Author Response · Authors · 2024-11-21
> **Rebuttal (1/4)**
>
> Thank you very much for your thoughtful and constructive feedback on our paper. We sincerely appreciate the time and effort you dedicated to reviewing our work, and will revise our paper carefully according to your valuable suggestions. Below, we will respond to each of your points in detail.
>
> ## Response to Weaknesses1
>
> - First, employing the method for measuring feature smoothness described in [1], we define the smoothness of temporal features near the boundaries of adjacent action segments (within $±T_b$ frames) as $\mathrm{Smooth}@T_b$. A higher level of smoothness corresponds to a lower $\mathrm{Smooth}@T_b$ value. The calculation formula is as follows: $\mathrm{Smooth}@T_b = {\frac{1}{T_b-1}}\sum\limits_{t=1}^{T_b-1}{\frac{1}{C}}\sum\limits_{c=1}^{C}|x_{t+1,c}-x_{t,c}|.$
>
>   We evaluated the smoothness $\mathrm{Smooth}@\{5,10,20\}$ of temporal features extracted by FACT and our proposed MFAI on the PKU-MMD (X-sub) dataset. The results are shown in the following table. Additionally, since the effect of blurred action boundaries is reflected in the F1 scores, we also provide the F1 scores for both methods on the PKU-MMD(X-sub) dataset.
>
>   |   Method   | Smooth@5  | Smooth@10 | Smooth@20 |  F1@10   |  F1@25   |  F1@50   |
>   | :--------: | :-------: | :-------: | :-------: | :------: | :------: | :------: |
>   |    FACT    |   0.021   |   0.020   |   0.018   |   76.0   |   71.4   |   56.6   |
>   | MFAI(Ours) | **0.069** | **0.064** | **0.058** | **82.8** | **80.4** | **69.9** |
>
>   It can be seen that the $\mathrm{Smooth}@\{5,10,20\}$ values of FACT are significantly lower than those of our MFAI, which implies that the temporal features extracted by FACT are notably smoother near the action boundaries compared to ours. Furthermore, the F1 scores of FACT also decrease compared to our MFAI, with a notable decrease of 13.3% in F1@50. These results confirm the existence of a relationship between over-smoothing features and blurred action boundaries.
>
> - Next, we will further explain the relationship between over-smoothing features and blurred action boundaries:
>
>   - Why does the problem of over-smoothing features frequently arise? In action sequences, transitions between different actions are gradual and smooth near the action boundaries. Additionally, the nature of optical flow data, which is used during the extraction of I3D features, promotes a tendency for these features to change smoothly near action boundaries. These factors lead to the temporal features extracted by the model tending to be over-smoothing near action boundaries.
>   - How does this smoothness contribute to blurred action boundaries? While the features exhibit smoothness near action boundaries, the action labels change abruptly, transitioning sharply from one action category to another. This disparity between the smooth feature transitions and abrupt label changes challenges the model's ability to pinpoint accurate action boundary locations, thus leading to the issue of blurred action boundaries.
>
> - The visualization of temporal feature embeddings generated by FACT, as shown in Figure 4(a), demonstrates that features from different action categories, distinguished by various colors, connect smoothly at the category boundaries. This phenomenon indicates a convergence of feature representations near the boundaries, resulting in smoother transitions of features near the action boundaries. Consequently, this exacerbates the problem of blurred action boundaries described above.
>
> - In Table 6, we present the comparative results of multi-scale frame-action interaction modeling with varying numbers of stages. Increasing $H$ from 0 to 3 significantly improves the performance. This is mainly due to the incorporation of more comprehensive temporal semantic information through multi-scale frame-action interaction modeling. This enhancement allows the model to better identify subtle differences between action categories and improve the discriminability of features near action boundaries. Consequently, these adjustments help mitigate the issue of over-smoothing features and lead to more separable category boundaries, which is evident from the improved F1 scores. However, when $H$ increases beyond 3, we can see that the performance starts to degrade. This is due to the over-fitting problem caused by the limited data volume and the increased number of model parameters.
>
>
>
> References:
>
> [1] Yifan H, Jian Z, James C, et al. Measuring and improving the use of graph information in graph neural network[C]//The Eighth International Conference on Learning Representations (ICLR 2020), Addis Ababa. 2020.

---

> > ### Author Response · Authors · 2024-11-21
> > **Rebuttal (2/4)**
> >
> > ## Response to Weaknesses2
> >
> > - We acknowledge that multi-scale temporal modeling and query-based approaches have been explored in previous works. However, it is important to note that our goal and methodology of employing multi-scale temporal modeling differ from those in prior studies (e.g., C2F-TCN [1] mentioned by you). Previous works, in order to mitigate the over-segmentation error in predictions, perform a single frame-level modeling at different temporal resolutions and generate the final prediction by implicitly integrating between temporal features at different time scales. However, relying solely on frame-level modeling often struggles to capture long-range dependencies in action sequences.
> >
> >   In contrast, to address the issue of over-smoothing features associated with frame-action modeling at fixed temporal resolution, we propose a multi-scale temporal modeling approach that performs frame-action interaction modeling at multiple temporal resolutions. As detailed in lines 279-282, action-level modeling at different time scales enables our model to effectively capture long-range dependencies within action sequences. Moreover, the bidirectional transfer of frame-level and action-level information across these scales allows the model to integrate the low-level detail information of frame-level features with the high-level dependencies of action-level features more effectively.  This integration enhances the representation of temporal semantics across various action categories, substantially reducing the problem of over-smoothing features.
> >
> > - Regarding the query-based method UVAST [2] you mentioned, as we described in Section 2.1 (lines 130-134), it utilizes a two-stage approach: the model initially learns frame-level features and makes preliminary predictions, and then constructs action-level features on top of them and further refines the predictions. However, this approach does not fully utilize the complementary information between frame-level features and action-level features, resulting in limited performance.
> >
> >   In contrast, our InterAct optimally utilizes the complementary information between frame-level and action-level information by performing bi-directional transfer across different temporal scales. This strategy allows the model to understand various temporal semantics within action sequences and to capture the temporal dynamics in action sequences more effectively, thus enhancing the temporal modeling capability.
> >
> > ## Response to Weaknesses3
> >
> > As described in the Introduction (lines 97-102) and Section 4.5 (lines 472-479), we propose the DSMI (Decoupled Spatial Modality Interaction) module to address the misalignment problem between spatial features and temporal modules. By employing DSMI, we can leverage the complementary information between different spatial modalities more comprehensively. Through interactive fusion between them, our extracted spatial features can better characterize the complex spatial relationships between different joints, resulting in stronger discriminative properties than those extracted by previous methods (as shown in Figure 3). This enhancement ensures that the spatial features better align with the temporal modules, which rely on the high discriminative properties of the I3D features.
> >
> >
> >
> > References:
> >
> > [1] Singhania D, Rahaman R, Yao A. C2F-TCN: A framework for semi-and fully-supervised temporal action segmentation[J]. IEEE Transactions on Pattern Analysis and Machine Intelligence, 2023, 45(10): 11484-11501.
> >
> > [2] Behrmann N, Golestaneh S A, Kolter Z, et al. Unified fully and timestamp supervised temporal action segmentation via sequence to sequence translation[C]//European conference on computer vision. Cham: Springer Nature Switzerland, 2022: 52-68.

---

> > > ### Author Response · Authors · 2024-11-21
> > > **Rebuttal (3/4)**
> > >
> > > ## Response to Weaknesses4
> > >
> > > - We appreciate this valuable suggestion. To maintain consistency in our ablation studies, we have conducted additional module ablation experiments on the PKU-MMD (X-sub) dataset. The results are shown below:
> > >
> > >   | Baseline | DSMI | MFAI |  F1@10   |  F1@25   |  F1@50   |   Edit   |   Acc    |
> > >   | :------: | :--: | :--: | :------: | :------: | :------: | :------: | :------: |
> > >   |    √     |      |      |   78.8   |   76.9   |   65.7   |   72.8   |   76.0   |
> > >   |    √     |  √   |      |   83.1   |   80.7   |   70.9   |   77.8   |   79.6   |
> > >   |    √     |      |  √   |   82.8   |   80.4   |   69.9   |   77.1   |   77.9   |
> > >   |    √     |  √   |  √   | **83.5** | **81.7** | **73.0** | **78.2** | **79.9** |
> > >
> > > - In addition, to evaluate the robustness of our approach, we also performed ablation experiments for RGB-based TAS. We conducted module ablations on the large RGB action segmentation dataset, Breakfast, that you mentioned. During the feature extraction phase, following prior studies,  we used a pretrained model to extract I3D features. Therefore, our ablation studies focus solely on the temporal module (i.e., MFAI). In the baseline setting, the model relies solely on frame-level temporal modeling. The results are shown in the following table:
> > >
> > >   | Baseline | MFAI |  F1@10   |  F1@25   |  F1@50   |   Edit   |   Acc    |
> > >   | :------: | :--: | :------: | :------: | :------: | :------: | :------: |
> > >   |    √     |      |   81.1   |   76.8   |   67.0   |   78.8   |   77.7   |
> > >   |    √     |  √   | **82.3** | **78.1** | **68.3** | **80.2** | **78.2** |
> > >
> > > ## Response to Weaknesses5
> > >
> > > Thanks for your insightful comment. We agree that reproducing results using the code provided by the authors of DiffAct [1], UVAST [2], and LTContext [3] on the MCFS-130 dataset would offer a more thorough comparison.
> > >
> > > Notably, DiffAct is Diffusion-based, UVAST is Query-based, and LTContext is Transformer-based. Given that both UVAST and ASQuery [4] are Query-based methods, and both LTContext and DSTN [5] are Transformer-based methods, with ASQuery and DSTN demonstrating superior performance on the challenging Breakfast dataset (as shown in the table below), we have included their results on MCFS-130 for a comparative analysis.
> > >
> > > |            Method             |  F1@10   |  F1@25   |  F1@50   |   Edit   |   Acc    |
> > > | :---------------------------: | :------: | :------: | :------: | :------: | :------: |
> > > | LTContext (Transformer-based) |   77.6   |   72.6   |   60.1   |   77.0   |   74.2   |
> > > |   DSTN (Transformer-based)    |   80.4   |   75.7   |   64.7   |   78.0   |   73.7   |
> > > |      UVAST (Query-based)      |   76.9   |   71.5   |   58.0   |   77.1   |   69.7   |
> > > |     ASQuery (Query-based)     |   80.7   |   76.5   |   66.5   |   78.4   |   77.9   |
> > > |   DiffAct (Diffusion-based)   |   80.3   |   75.9   |   64.6   |   78.4   |   76.4   |
> > > |        InterAct (Ours)        | **82.3** | **78.1** | **68.3** | **80.2** | **78.2** |
> > >
> > > To provide a more complete comparison, we will do our best to report the results for the methods you mentioned. Unfortunately, since MCFS-130 is a large dataset, these experiments require time, preventing us from including all results in this response. We first reproduced the Diffusion-based DiffAct to cover all types of methods. The results of its comparison with our InterAct on MCFS-130 (RGB) are shown in the table below.
> > >
> > > |     Method      |  F1@10   |  F1@25   |  F1@50   |   Edit   |   Acc    |
> > > | :-------------: | :------: | :------: | :------: | :------: | :------: |
> > > |     DiffAct     |   53.2   |   48.9   |   36.4   |   54.1   |   59.5   |
> > > | InterAct (Ours) | **58.9** | **55.4** | **45.9** | **57.3** | **62.2** |
> > >
> > > Regardless, if the paper is accepted, we will expand the final version to include the reproduction results of these methods to provide a more complete comparison.
> > >
> > >
> > >
> > > References:
> > >
> > > [1] Liu D, Li Q, Dinh A D, et al. Diffusion action segmentation[C]//Proceedings of the IEEE/CVF International Conference on Computer Vision. 2023: 10139-10149.
> > >
> > > [2] Behrmann N, Golestaneh S A, Kolter Z, et al. Unified fully and timestamp supervised temporal action segmentation via sequence to sequence translation[C]//European conference on computer vision. Cham: Springer Nature Switzerland, 2022: 52-68.
> > >
> > > [3] Bahrami E, Francesca G, Gall J. How Much Temporal Long-Term Context is Needed for Action Segmentation?[C]//Proceedings of the IEEE/CVF International Conference on Computer Vision. 2023: 10351-10361.
> > >
> > > [4] Gan Z, Jin L, Nie L, et al. ASQuery: A Query-based Model for Action Segmentation[C]//2024 IEEE International Conference on Multimedia and Expo (ICME). IEEE, 2024: i-vi.
> > >
> > > [5] Duan H, Liu S, Tan C, et al. Decoupling Spatio-Temporal Network for Fine-Grained Temporal Action Segmentation[C]//2024 IEEE International Conference on Multimedia and Expo (ICME). IEEE, 2024: 1-6.

---

> > > > ### Author Response · Authors · 2024-11-21
> > > > **Rebuttal (4/4)**
> > > >
> > > > ## Response to Weaknesses6
> > > >
> > > > Thank you for your constructive comments. We have further conducted ablation studies on the number of layers $L$ in the encoder and decoder on the PKU-MMD (X-sub) dataset. The results are as follows:
> > > >
> > > > | $L$  |  F1@10   |  F1@25   |  F1@50   |   Edit   |   Acc    |   Avg    |
> > > > | :--: | :------: | :------: | :------: | :------: | :------: | :------: |
> > > > |  2   |   78.5   |   75.6   |   65.6   |   72.1   |   74.1   |   73.2   |
> > > > |  4   |   82.9   |   80.9   |   71.5   |   77.8   |   79.0   |   78.4   |
> > > > |  6   |   83.5   | **81.7** | **73.0** |   78.2   | **79.9** | **79.3** |
> > > > |  8   |   83.5   |   81.6   |   72.3   |   78.2   |   78.7   |   78.9   |
> > > > |  10  | **83.7** | **81.7** |   72.9   | **78.3** |   78.8   |   79.1   |
> > > >
> > > > It is observed that increasing $L$ from 2 to 6 greatly boosts the performance, highlighting the importance of sufficient temporal granularity. Further increasing $L$ to 8 or 10 still improves the performance, but the improvements are less significant. We set $L$ = 6 as it provides the best overall performance across all evaluated metrics.
> > > >
> > > >
> > > >
> > > > Once again, we appreciate your thoughtful feedback and hope our responses address your concerns. We will incorporate the suggested revisions to improve the clarity and completeness of our paper.

---

### Official Review · Reviewer_hFEf · 2024-11-02

**Soundness:** 2
**Presentation:** 2
**Contribution:** 2
**Rating:** 5
**Confidence:** 5

**Summary:**

To enhance skeleton-based action segmentation by extracting more discriminative features, the authors propose multi-scale spatial modeling to fuse different modalities. To improve action segmentation by capturing complex temporal dynamics, the authors propose a MFAI that facilitates frame-action interactions across multiple temporal scales.

**Strengths:**

This paper clearly outlines its motivations, proposing a MFAI for frame-action interactions and a DSMI for fusing spatial features, with a certain degree of originality.

**Weaknesses:**

1.	The motivations outlined in the Introduction assume that iterative frame-action interactions at a constant temporal resolution would result in over-smoothing. To substantiate this, a visualization experiment is conducted, which shows that boundaries become more visually discernible. However, the effect may be due to the boundary prediction module and the existing label smoothing method. When aiming to elucidate the precise role of a structure, it is advisable to conduct a single-variable analysis. Regrettably, the paper lacks ablation experiments about both the boundary prediction module and the operations at a fixed temporal resolution. Therefore, the experimental support is insufficient for the conclusion, and the analysis is not rigorous, which makes the motivation speculative and lacks persuasiveness. Suggestions:
(1) Conduct a comparative analysis of t-SNE visualization results, where one is derived from InterAct and another is from a modified version of InterAct devoid of the BPM.
(2) To ascertain whether a fixed temporal resolution poses a limitation, compare t-SNE visualization outcomes between InterAct and a variant where the encoder-decoder is substituted with MS-TCN.
(3) To evaluate the impact of label smoothing, eliminate the BPM and replace label smoothing with the Average employed by FACT.

2.	Some SOTA methods are not compared in this paper, such as ASPnet, Semantic2Graph, and BIT. I suggest not to ignore these methods that perform better than the proposed one, because some methods also adopt frame-action interaction, such as BIT, and multiply modalities fusion, such as ASPnet. Comparing your method with the above SOTA methods can effectively underscore the merits of your solution, particularly because: (1) Your DSMI serves as a fusion strategy, echoing similar strategies proposed by ASPNet and Semantic2Graph in feature fusion. (2) Your MFAI constitutes a frame-action interaction module, paralleling the frame-action interaction module introduced by BIT.

3.	Many grammar errors need to be corrected, for example,
    (1) The last line of page 4, “suffers from” should be “suffer from”.
    (2) In Section 2.1, “frame and action level” should be revised to “frame and action levels”.
    (3) In the line before equation (8), “as follow” should be revised to “as follows”.
    (4) Line -5 of page 5, “D is the feature dimensions” should be revised.
We suggest that the authors have the paper professionally proofread or reviewed by a native English speaker to address language issues comprehensively.

4.	Both MFAI and MAFI are used in this paper, which makes me confused about the paper. We suggest consistently use one acronym throughout the paper and include it in a list of abbreviations for clarity.

**Questions:**

Did you experiment with replacing the encoder-decoder with models like MS-TCN or ASFormer that operate on fixed full temporal resolution? If yes, please add the descriptions. If not, your motivation for multi-scale analysis in the introduction lacks support.

---

> ### Author Response · Authors · 2024-11-21
> **Rebuttal (1/2)**
>
> Thank you for your insightful review of our manuscript. We greatly appreciate the thoroughness of your comments and the helpful suggestions, which will certainly strengthen our paper. Below, we provide our responses to the points you have kindly raised.
>
> ## Response to Weaknesses1
>
> Thanks for your constructive comments. We acknowledge that incorporating the ablation experiments you suggested to add could significantly enhance the rigor of our work. Therefore, based on your suggestion, we have conducted additional ablation studies on the effects of boundary prediction module, the label smoothing method, and fixed temporal resolution modeling on temporal feature smoothness.
>
> First, employing the method for measuring feature smoothness described in [1], we define the smoothness of temporal features near the boundaries of adjacent action segments (within $±T_b$ frames) as $\mathrm{Smooth}@T_b$. A higher level of smoothness corresponds to a lower $\mathrm{Smooth}@T_b$ value. The calculation formula is as follows: $\mathrm{Smooth}@T_b = {\frac{1}{T_b-1}}\sum\limits_{t=1}^{T_b-1}{\frac{1}{C}}\sum\limits_{c=1}^{C}|x_{t+1,c}-x_{t,c}|.$
>
> We evaluated the smoothness $\mathrm{Smooth}@\{5,10,20\}$ of temporal features extracted by various methods on the PKU-MMD (X-sub) dataset. The results are shown in the following table. Additionally, since over-smoothing features lead to blurred action boundaries, which is reflected in the F1 scores, we also provide the F1 scores of these methods on the PKU-MMD (X-sub) dataset. In the table, $'$ denotes replacing the encoder-decoder with MS-TCN, $^+$ denotes removing the boundary prediction module, and $^*$ denotes removing the boundary prediction module and replacing the label smoothing method with the Average adopted by FACT [2].
>
> |    Method    | $\mathrm{Smooth}@5$ | $\mathrm{Smooth}@10$ | $\mathrm{Smooth}@20$ |  F1@10   |  F1@25   |  F1@50   |
> | :----------: | :-----------------: | :------------------: | :------------------: | :------: | :------: | :------: |
> |     FACT     |        0.021        |        0.020         |        0.018         |   76.0   |   71.4   |   56.6   |
> | InterAct$'$  |        0.044        |        0.041         |        0.036         |   78.7   |   73.9   |   60.5   |
> | InterAct$^+$ |        0.061        |        0.056         |        0.052         |   80.1   |   75.7   |   64.9   |
> | InterAct$^*$ |        0.056        |        0.052         |        0.047         |   79.2   |   74.4   |   62.8   |
> |   InterAct   |      **0.069**      |      **0.064**       |      **0.058**       | **82.8** | **80.4** | **69.9** |
>
> - After replacing the encoder-decoder with MS-TCN, the $\mathrm{Smooth}@\{5, 10, 20\}$ values are significantly reduced, with a notable decrease of 2.5% in $\mathrm{Smooth}@5$. This suggests that temporal features are significantly smoother near the action boundaries. This implies that iterative frame-action interaction modeling at fixed temporal resolution is an important factor leading to over-smoothing features.
>
>   In contrast, removing the boundary prediction module and replacing the label smoothing method have a significantly lesser impact on feature smoothness. These two components were originally designed to mitigate over-segmentation errors, so they do have some impact on the discriminative properties of features near action boundaries. But the iterative frame-action interaction modeling at a fixed temporal resolution still remains the dominant factor influencing feature smoothness.
>
> - After replacing the encoder-decoder with MS-TCN, F1@50 decreases from 69.9 to 60.5. Although the removal of the boundary prediction module and the alteration of the label smoothing method also impact the accuracy of boundary predictions, their effects are considerably less significant than the replacement of the encoder-decoder with MS-TCN. This further underscores that frame-action interaction modeling with a fixed temporal resolution is the main driver behind the over-smoothing features.
>
> If the paper is accepted, we will expand the content in the final version to include visualization of the comparative results of these ablation experiments to provide a more complete comparison. Thanks again for your insightful comments and suggestions.
>
>
>
> References:
>
> [1] Yifan H, Jian Z, James C, et al. Measuring and improving the use of graph information in graph neural network[C]//The Eighth International Conference on Learning Representations (ICLR 2020), Addis Ababa. 2020.
>
> [2] Lu Z, Elhamifar E. Fact: Frame-action cross-attention temporal modeling for efficient action segmentation[C]//Proceedings of the IEEE/CVF Conference on Computer Vision and Pattern Recognition. 2024: 18175-18185.

---

> > ### Author Response · Authors · 2024-11-21
> > **Rebuttal (2/2)**
> >
> > ## Response to Weaknesses2
> >
> > - To ensure a fair comparison, we initially did not compare some of the recent SOTA methods such as Br-Prompt [1], Semantic2Graph [2], and ASPnet [3]. Because these methods utilize additional multimodal features, making them unsuitable for direct comparison.
> >   However, it is noteworthy that on the Breakfast dataset, a large and challenging dataset for RGB-based Temporal Action Segmentation (TAS), our InterAct significantly outperforms ASPnet when ASPnet does not use additional modal features (i.e., using only RGB video data), as shown in the table below：
> >
> >   |     Method      |  F1@10   |  F1@25   |  F1@50   |   Edit   |   Acc    |
> >   | :-------------: | :------: | :------: | :------: | :------: | :------: |
> >   |     ASPnet      |   78.1   |   72.9   |   60.8   |   76.3   |   75.9   |
> >   | InterAct (Ours) | **82.3** | **78.1** | **68.3** | **80.2** | **78.2** |
> >
> > - As for the BIT [4] you mentioned, to our understanding, it refers to the same method as FACT [5] discussed in this paper, simply renamed.
> >
> > ## Response to Weaknesses3 and 4
> >
> > We sincerely apologize for the grammatical and acronym-related errors. Thank you very much for your suggestions. We have proofread the article more carefully and will address the language and acronym issues in the next version.
> >
> >
> >
> > References:
> >
> > [1] Li M, Chen L, Duan Y, et al. Bridge-prompt: Towards ordinal action understanding in instructional videos[C]//Proceedings of the IEEE/CVF conference on computer vision and pattern recognition. 2022: 19880-19889.
> >
> > [2] Zhang J, Tsai P H, Tsai M H. Semantic2Graph: graph-based multi-modal feature fusion for action segmentation in videos[J]. Applied Intelligence, 2024, 54(2): 2084-2099.
> >
> > [3] van Amsterdam B, Kadkhodamohammadi A, Luengo I, et al. Aspnet: Action segmentation with shared-private representation of multiple data sources[C]//Proceedings of the IEEE/CVF Conference on Computer Vision and Pattern Recognition. 2023: 2384-2393.
> >
> > [4] Lu Z, Elhamifar E. BIT: Bi-Level Temporal Modeling for Efficient Supervised Action Segmentation[J]. arXiv preprint arXiv:2308.14900, 2023.
> >
> > [5] Lu Z, Elhamifar E. Fact: Frame-action cross-attention temporal modeling for efficient action segmentation[C]//Proceedings of the IEEE/CVF Conference on Computer Vision and Pattern Recognition. 2024: 18175-18185.

---

> ### Comment · Reviewer_hFEf · 2024-11-24
>
> Thanks for your responses. For response 1, the augmented experimental results show the impact of fixed temporal resolution modeling on feature smoothness, thereby it can reflect the support for the pertinent assertions made in the Introduction. For response 2, the rationale provided for their initial comparison choices is justifiable. For 3 and 4, the authors’ stance in addressing and correcting errors is proactive. In short, I'm satisfied with the authors' reply, so I increased my score accordingly.

---

### Official Review · Reviewer_mM7P · 2024-11-08

**Soundness:** 2
**Presentation:** 3
**Contribution:** 2
**Rating:** 5
**Confidence:** 4

**Summary:**

This paper proposes a novel Interaction-based framework for Action segmentation,  which integrates multiple temporal resolutions
for frame-action modeling, thereby enhancing temporal interactions. Extensive experimental results demonstrate its effectiveness.

**Strengths:**

This paper is well-written and it achieves good performance on the benchmarks.

**Weaknesses:**

- The contribution of multi-scale temporal modeling have been studied well by many previous works. So this paper has limited novelty.
- Missing recent SOTA methods, like Semantic2Graph and ASPnet.
- Whether the two modal data (RGB and skeleton data) fusion can be used as the input of the model, the experimental results can provide help to the author.

**Questions:**

See Weaknesses.

---

> ### Author Response · Authors · 2024-11-21
> **Rebuttal**
>
> Thanks for your time and efforts in reviewing our paper! We highly appreciate your thoughtful and constructive suggestions. Your thoughtful and constructive suggestions have been invaluable to us, and we have carefully considered each comment. Our responses to your queries are outlined below:
>
> ## Response to Weaknesses1
>
> - We acknowledge that multi-scale temporal modeling is a widely adopted technique in the field. However, it is important to note that our goal and methodology of employing multi-scale temporal modeling differ from those in prior studies (e.g., C2F-TCN [1] mentioned by reviewer iX6Z). Previous works, in order to mitigate the over-segmentation error in predictions, perform a single frame-level modeling at different temporal resolutions and generate the final prediction by implicitly integrating between temporal features at different time scales. However, relying solely on frame-level modeling often struggles to capture long-range dependencies in action sequences.
> - In contrast, to address the issue of over-smoothing features associated with frame-action modeling at fixed temporal resolution, we propose a multi-scale temporal modeling approach that performs frame-action interaction modeling at multiple temporal resolutions. As detailed in lines 279-282, action-level modeling at different time scales enables our model to effectively capture long-range dependencies within action sequences. Moreover, the bidirectional transfer of frame-level and action-level information across these scales allows the model to integrate the low-level detail information of frame-level features with the high-level dependencies of action-level features more effectively.  This integration enhances the representation of temporal semantics across various action categories, substantially reducing the problem of over-smoothing features.
>
> ## Response to Weaknesses2
>
> To ensure a fair comparison, we initially did not compare some of the recent SOTA methods such as Br-Prompt [2], Semantic2Graph [3], and ASPnet [4]. Because these methods utilize additional multimodal features, making them unsuitable for direct comparison.
> However, it is noteworthy that on the Breakfast dataset, a large and challenging dataset for RGB-based Temporal Action Segmentation (TAS), our InterAct significantly outperforms ASPnet when ASPnet does not use additional modal features (i.e., using only RGB video data), as shown in the table below：
>
> |     Method      |  F1@10   |  F1@25   |  F1@50   |   Edit   |   Acc    |
> | :-------------: | :------: | :------: | :------: | :------: | :------: |
> |     ASPnet      |   78.1   |   72.9   |   60.8   |   76.3   |   75.9   |
> | InterAct (Ours) | **82.3** | **78.1** | **68.3** | **80.2** | **78.2** |
>
> ## Response to Weaknesses3
>
> In this work, we mainly focus on designing a unified framework to address the two tasks of skeleton-based TAS and RGB-based TAS simultaneously. Considering the inherent heterogeneity of different modalities (e.g., scale, resolution, and dynamic range) and the computational challenges associated with processing and integrating large-scale multimodal data in practical scenarios, we opted not to design our model to incorporate RGB, skeleton, or other types of modal data (e.g., text) as inputs. However, we acknowledge that the idea of using data fusion of the two modalities as model input is promising and we will consider this in future work.
>
>
>
> References:
>
> [1] Singhania D, Rahaman R, Yao A. C2F-TCN: A framework for semi-and fully-supervised temporal action segmentation[J]. IEEE Transactions on Pattern Analysis and Machine Intelligence, 2023, 45(10): 11484-11501.
>
> [2] Li M, Chen L, Duan Y, et al. Bridge-prompt: Towards ordinal action understanding in instructional videos[C]//Proceedings of the IEEE/CVF conference on computer vision and pattern recognition. 2022: 19880-19889.
>
> [3] Zhang J, Tsai P H, Tsai M H. Semantic2Graph: graph-based multi-modal feature fusion for action segmentation in videos[J]. Applied Intelligence, 2024, 54(2): 2084-2099.
>
> [4] van Amsterdam B, Kadkhodamohammadi A, Luengo I, et al. Aspnet: Action segmentation with shared-private representation of multiple data sources[C]//Proceedings of the IEEE/CVF Conference on Computer Vision and Pattern Recognition. 2023: 2384-2393.

---

### Meta-Review · Area_Chair_7Ef2 · 2024-12-16

**Metareview:**

This paper clearly outlines its motivations, proposing a MFAI for frame-action interactions and a DSMI for fusing spatial features, with a certain degree of originality. However, reviewers pointed out that this paper lacked necessary experimental results and ablation experiments, which did not fully address the above concerns after rebuttal. The final vote is Reject.

**Additional Comments On Reviewer Discussion:**

Limited novelty and missing the necessary ablation experiments of the motivation and contributions.

---

### Decision · Program_Chairs · 2025-01-22

Reject